

# What do alexithymia items measure? A discriminant content validity study of the Toronto-alexithymia-scale–20

Elke Veirman[1,*], Dimitri M.L. Van Ryckeghem[1,2,3,*], Gregory Verleysen[4,5], Annick L. De Paepe[1] and Geert Crombez[1]

[1] Department of Experimental-Clinical and Health Psychology, Faculty of Psychology and Educational Sciences, Ghent University, Ghent, Belgium
[2] Department of Behavioural and Cognitive Sciences, Faculty of Humanities, Education and Social Sciences, University of Luxembourg, Esch-sur-Alzette, Luxembourg
[3] Experimental Health Psychology, Clinical Psychological Science, Faculty of Psychology and Neuroscience, Maastricht University, Maastricht, Netherlands
[4] Centre for Research and Innovation in Care, Faculty of Medicine and Health Sciences, University of Antwerp, Antwerp, Belgium
[5] End-of-Life Care Research group, Faculty of Medicine and Pharmacy, Vrije Universiteit Brussel, Brussels, Belgium
* These authors contributed equally to this work.

## ABSTRACT

**Background:** Questions have been raised about whether items of alexithymia scales assess the construct alexithymia and its key features, and no other related constructs. This study assessed the (discriminant) content validity of the most widely used alexithymia scale, i.e., the Toronto Alexithymia Scale (TAS-20).

**Methods:** Participants ($n$ = 81) rated to what extent TAS-20 items and items of related constructs were relevant for assessing the constructs 'alexithymia', 'difficulty identifying feelings', 'difficulty describing feelings', 'externally-oriented thinking', 'limited imaginal capacity', 'anxiety', 'depression', and 'health anxiety'.

**Results:** Results revealed that, overall, the TAS-20 did only partly measure 'alexithymia'. Only the subscales 'difficulty identifying feelings' and 'difficulty describing feelings' represented 'alexithymia' and their intended construct, although some content overlap between these subscales was found. In addition, some items assessed (health) anxiety equally well or even better.

**Conclusions:** Revision of the TAS-20 is recommended to adequately assess all key features of alexithymia. Findings with the TAS-20 need to be interpreted with caution in people suffering from medical conditions.

Corresponding author
Elke Veirman,
elke.veirman@ugent.be

## INTRODUCTION

The alexithymia construct has been introduced in the early seventies by *Sifneos (1972, 1973)* to describe clinical observations of patients with classic psychosomatic diseases who had difficulty engaging in insight-oriented psychotherapy (e.g., *MacLean, 1949*; *Marty & de M'Uzan, 1963*; *Ruesch, 1948*; *Sifneos, 1967*). Since then alexithymia, defined as the inability to recognize and express emotions (*Taylor, Bagby & Parker, 2016*), has been

considered a key construct in many theoretical models of health psychology (*Lumley, Neely & Burger, 2007*). Contemporary theories describe alexithymia as a multidimensional construct with four interrelated features: *"(1) difficulty identifying feelings and distinguishing between feelings and the bodily sensations of emotional arousal, (2) difficulty describing feelings to other people, (3) constricted imaginal processes, as evidenced by a paucity of fantasies, and (4) a stimulus-bound, externally oriented cognitive style"* (*Taylor, Bagby & Parker, 1997*, p. 29; see also *Sifneos, 1994*). Over the last decades, alexithymia has been recognized as a risk factor for various psychiatric and medical conditions (*Corcos & Speranza, 2003*; *Taylor, Bagby & Parker, 1997*). In particular, it has been theorized that alexithymia reflects a deficit in the cognitive processing and regulation of emotions (*Taylor, Bagby & Parker, 1997*). This deficit would increase one's vulnerability for psychiatric and medical conditions (*Fernandez, Jazaieri & Gross, 2016*). This idea is furthermore supported by abundant research showing that levels of alexithymia are increased in patients suffering from illnesses, such as eating disorders (e.g., *Taylor et al., 1996*), posttraumatic stress disorders (e.g., *Frewen et al., 2006*), chronic pain (e.g., *Pecukonis, 2009*), cancer (e.g., *Todarello et al., 1989*), and many more (*Luminet, Bagby & Taylor, 2018*; for a review, see *Taylor & Bagby, 2000*; *Taylor, 2004*).

The Toronto Alexithymia Scale–20 (TAS-20; *Bagby, Parker & Taylor, 1994a*; *Bagby, Taylor & Parker, 1994b*) is worldwide the most frequently used measure of alexithymia in both research and clinical practice (*Lane et al., 2015*; *Sekely, Bagby & Porcelli, 2018*). Although the TAS-20 is considered to be a well validated self-report measure of alexithymia (e.g., *Bagby, Parker & Taylor, 2020*), some concerns about its validity remain (*Bermond, Oosterveld & Vorst, 2015*; *Lane et al., 2015*; *Lumley, Neely & Burger, 2007*). First, doubts have been raised about whether the TAS-20 measures alexithymia in a comprehensive and relevant manner. The TAS-20 contains three subscales, i.e., 'difficulty identifying feelings', 'difficulty describing feelings', and 'externally-oriented thinking'. The items for assessing the daydreaming factor in the earlier revision of the original TAS (TAS-R; *Taylor, Ryan & Bagby, 1985*; *Taylor, Bagby & Parker, 1992*), were eliminated because of either low item-total correlations or high correlations with a social desirability measure (*Bagby, Parker & Taylor, 1994a*; *Bagby, Taylor & Parker, 1994b*). *Bagby, Parker & Taylor (1994a*, *1994b*, see also *Taylor, Bagby & Parker (2016))* motivated their decision by arguing that this factor may be indirectly measured by the factor 'externally-oriented thinking' (*Bagby, Parker & Taylor, 2020*). Furthermore, confirmatory and exploratory factor analyses show that at least half of the externally-oriented thinking items load poorly on their intended factor (factor loadings < 0.40; e.g., *Kooiman, Spinhoven & Trijsburg, 2002*; *Taylor, Bagby & Parker, 2003*). If aspects of the construct alexithymia are underrepresented by the TAS-20 items and/or TAS-20 items are not relevant for the construct, it reflects a lack of content validity.

Second, there are doubts about whether the TAS-20 is sufficiently distinct from measures assessing related theoretical constructs. Some authors have argued that the TAS-20 is a measure of psychological distress rather than alexithymia (*Leising, Grande & Faber,*

*2009*). Indeed, significant and substantial correlations have been reported between the TAS-20 and measures of anxiety and depression in clinical samples (e.g., *Marchesi et al., 2014*) and in the general population (e.g., *Honkalampi et al., 2010*). Furthermore, *Shahidi, Molaie & Dehghani (2012)* found significant correlations between the TAS-20 scores and a measure of health anxiety. This study revealed that the 'difficulty identifying feelings' subscale predicted 52% of the total variance in health anxiety scores, and argued that this strong relationship is driven by particular items that measure difficulty in differentiating between bodily feelings and emotions (see also *Barsky, 2001*; *De Gucht, Fischler & Heiser, 2004*; *Nakao et al., 2002*). It is key that correlations between the TAS-20 and health anxiety are not (partially) explained by content overlap. If the TAS-20 is contaminated by content relevant to related constructs such as anxiety, depression, and health anxiety, relationships between the measures of these constructs may then simply be due to content overlap resulting in inflated explanatory power of alexithymia and hazardous theory building (*Dixon & Johnston, 2019*).

Despite these concerns, no study has examined the *content validity* and *discriminant content validity* of the TAS-20. In the current study, TAS-20 items are evaluated using the Discriminant Content Validity method (DCV; *Johnston et al., 2014*), a systematic and transparent way of investigating and reporting whether items are relevant for measuring target theoretical constructs (a key feature of content validity) and whether items are distinct from the content from other theoretical constructs (discriminant content validity). More specifically, we investigated to what extent items from the TAS-20 are (a) relevant for the construct 'alexithymia', and its key features, i.e., 'difficulty identifying feelings', 'difficulty describing feelings', 'externally-oriented thinking', and 'limited imaginal capacity' (content validity), and (b) distinct from related constructs, i.e., 'anxiety', 'depression', and 'health anxiety' (discriminant content validity).

## MATERIALS & METHODS

### Participants

Participants were 81 psychology students (English track) recruited at Maastricht University via Sona Systems, a cloud-based participant pool management software package (https://maastricht-fpn.sona-systems.com). They were not experts in alexithymia research as we wanted people to decide whether an item assesses a construct based upon the item and construct definition without knowledge bias (i.e., a theoretical background in the field of alexithymia). Data from participants were only included for the statistical analysis when participants were able to complete the online assessment in line with given instructions and quality checks (performance criteria).

### Discriminant content validity method

The Discriminant Content Validity method (DCV) method is a quantitative procedure to assess the (discriminant) content of theory-based measures (for a detailed overview of the

methodology, see *Johnston et al., 2014*). Here, we describe the DCV questionnaire we developed in five steps:

### Step 1: identification of constructs

Eight constructs were identified to be used for the categorization of the items. These constructs were 'alexithymia', 'difficulty identifying feelings', 'difficulty describing feelings', 'externally-oriented thinking', 'limited imaginal capacity', 'anxiety', 'depression', and 'health anxiety'. The constructs 'alexithymia', 'difficulty identifying feelings', 'difficulty describing feelings', 'externally-oriented thinking', and 'limited imaginal capacity' were selected to investigate to what extent TAS-20 items are identified as items that assess alexithymia, and to what extent they are identified to assess the respective key features of alexithymia (content validity). The constructs 'anxiety', 'depression', and 'health anxiety' were selected to investigate to what extent the TAS-20 item-content could be differentiated from other constructs to which alexithymia has been related (discriminant content validity). Finally, an 'other' category was added, preventing the impression that all items had to be categorized as measures of one of the predefined constructs.

### Step 2: construct definitions

Definitions were formulated for each of the identified constructs. The definition of alexithymia was based upon the definition of alexithymia provided by the online Oxford Living Dictionaries for English (https://en.oxforddictionaries.com accessed on 11/10/2018). This definition is a representation of how the construct is understood in lay terms and also corresponds to the scientific definition that is widely accepted (*Taylor, Bagby & Parker, 2016*). For the alexithymia features, definitions were based upon the widely acknowledged definitions of *Taylor, Bagby & Parker (1997)*. For the other predefined constructs, there are multiple definitions available, which could introduce bias in our findings due to preferring the definition of one theoretical framework over another. Therefore, we opted to base our definitions on those provided by the Online Oxford Living Dictionaries for English (https://en.oxforddictionaries.com accessed on 11/10/2018). The following definitions were used: (1) alexithymia: '*The inability to recognize one's own emotions and to express them, especially in words*'; (2) difficulty identifying feelings: '*Difficulty identifying feelings and distinguishing between feelings and the bodily sensations of emotional arousal.*'; (3) difficulty describing feelings: '*Difficulty describing feelings to other people.*'; (4) externally-oriented thinking: '*A stimulus-bound, externally oriented cognitive style.*'; (5) limited imaginal capacity: '*Constricted imaginal processes, as evidenced by a paucity of fantasies.*'; (6) anxiety: '*A feeling of worry, nervousness, or unease about something with an uncertain outcome.*'; (7) depression: '*Feelings of severe despondency and dejection.*'; and (8) health anxiety: '*A feeling of worry, nervousness, or unease about one's health.*'

### Step 3: selection of alexithymia items

The TAS-20 comprises 20 items across three subscales, with most of the items positively keyed (+) and some negatively keyed (−): 'difficulty identifying feelings' (items 1+, 3+, 6+, 7+, 9+, 13+, and 14+; e.g., "I am often confused about what emotion I am feeling"), 'difficulty describing feelings' (items 2+, 4−, 11+, 12+, and 17+; e.g., "It is difficult for me to

find the right words for my feelings"), and 'externally-oriented thinking' (items 5−, 8+, 10−, 15+, 16+, 18−, 19−, and 20+; e.g., "I prefer to analyze problems rather than just describe them"). Items are displayed in Table S1.

### Step 4: selection of items for the other constructs

For 'anxiety', four items (e.g., "I felt fearful") were retrieved from the PROMIS® Item Bank v1.0-Emotional Distress-Anxiety–Short Form 4a (PROMIS-A; *Pilkonis et al., 2011*; Table S2). For 'depression', four items (e.g., "I felt hopeless") were retrieved from the PROMIS® Item Bank v1.0–Emotional Distress-Depression–Short Form 4a (PROMIS-D; *Pilkonis et al., 2011*; Table S3). For 'health anxiety', four items (e.g., "I usually think that I am seriously ill") were retrieved from the Short Health Anxiety Inventory (SHAI; *Salkovskis et al., 2002*; Table S4). For feasibility reasons (i.e., reducing fatigue effects), the number of items for each contrast construct was limited to four.

### Step 5: rating scale of items

Participants were instructed to rate two questions per construct for each item (e.g., *Johnston et al., 2014*). In the first question, participants were asked to judge whether an item assesses a particular construct (common-scored items: 'no' and 'yes when reverse scored' = −1, whereas 'yes' = 1; reverse-scored items: 'no' and 'yes' = −1, whereas 'yes when reverse scored' = 1). In the second question, participants were asked to indicate on an 11 point scale (0 = *0% confidence* to 10 = *100% confidence*) to what extent they were confident about their judgment. Weighted judgements were calculated to express the relationship between each item and each construct. The code of the answer for 'no', 'yes', and 'yes when reverse scored' was multiplied with its accompanied confidence score, resulting in an *outcome score* with values ranging from −10 to +10.

## Self-report measures

### Participant characteristics

After completion of the DCV items, participants were asked to provide demographic information including gender, age, nationality, ethnicity, and current health status.

### PROMIS health profile

To provide information on the physical and mental health of the participants, the PROMIS® Profile-v2.1-PROMIS-29 was filled out, which contains seven scales, i.e., physical function (4 items), anxiety (4 items), depression (4 items), fatigue (4 items), sleep disturbance (4 items), ability to participate in social roles and activities (4 items), pain interference (4 items), and a pain intensity item. All items, except for the pain intensity item, are scored on a 5-point Likert scale. The pain intensity item "In the last 7 days, how would you rate your pain on average?" is rated on a 11-point Likert scale ranging from 0 (*no pain*) to 10 (*worst imaginable pain*) (*Hays et al., 1994*). Scale summary scores are transformed into standardized T-scores with a mean of 50 and a standard deviation (SD) of 10. Higher scores reflect more of the concept being measured. Research indicated that this questionnaire is reliable and valid for assessing health-related quality of life in the

general population and in populations with chronic health conditions (*Hays et al., 2018*; *Rose et al., 2018*).

### Detection of careless responding

Detection of careless responding (e.g., *Meade & Craig, 2012*; *Oppenheimer, Meyvis & Davidenko, 2009*) was built-in via three ways. First, the 'other' category provided the opportunity to check whether participants followed the given instructions. In particular, it was considered impossible for participants to provide the same extreme scores (i.e., −10 or +10) for an item on all predefined constructs and the 'other' category. Second, the DCV items were intermixed with three items from the Instructional Manipulation Check (IMC; e.g., "Please check yes and 30% for all constructs"). Third, an additional item was added at the end of the survey, asking participants how attentive they were when filling out the questionnaire (1 = *completely attentive*, 2 = *moderately attentive*, 3 = *not attentive at all*).

## Procedure

The study was approved by the Ethics Review Committee Psychology and Neuroscience (ERCPN) of Maastricht University, Maastricht, Netherlands (Ethical Application Ref: RP2027_2019_16). Questionnaires and DCV were assessed via an online survey constructed using Qualtrics ResearchCore™. Participants were invited at the university to participate in this study. Once seated, participants were welcomed by a researcher and received an information letter and signed a declaration of consent. Next, participants started the online assessment in a university room. Particularly, participants were provided with the instructions of the DCV method and one non-related example on how the DCV should be completed. After the instructions, participants were provided with one of two DCV item sets. Each DCV item set contained all items, but differed in the order in which the constructs had to be filled out (two random orders were drawn in advance which remained consistent throughout a person's assessment). The order in which the 35 DCV items (including three IMC items) were presented was random for each participant. After participants completed these DCV items, they provided demographic information, answered the additional question to detect careless responding, and filled out the questions assessing their physical and mental health (PROMIS® Profile-v2.1-PROMIS-29).

Finally, to reduce careless responding, each participant was forced to spend at least 30 s on each question to avoid quick and random answers. After finishing the survey, participants received an oral debriefing about the purpose of the study. The online assessment lasted on average 45.86 min (SD = 17.93 min). Participants received course credits for participation in the study.

## Analyses

Data collected with the DCV method were analyzed using Bayesian hierarchical models (JAGS version 4.3.0) in R version 3.6.0 (*R Core Team, 2019*). This methodology allows to perform analyses at measure as well as at item level, while ensuring that estimates do not fall outside the actual response range [−10 to +10] (see also *Crombez et al., 2020*). In the models a different mu parameter was estimated for each construct or measure, depending on the research question (see below). In addition, a random effect for subject and

item was added. All parameters received vague priors (normal distributions with a very large standard deviation; see *Crombez et al., 2020*). The dependent variable was the DCV *outcome score* (ranging from −10 to +10). The mu parameters come from a truncated normal distribution [−10, 10] so the credibility intervals only contain sensible values. To generate the posterior samples, we used four chains with 20,000 iterations each, 5,000 being discarded as burn in. Traceplots and Rhat values of 1 indicated that all the chains for the mu parameters reached convergence. The actual analyses were performed in three steps.

*First*, we investigated whether the items of the TAS-20, PROMIS-A, PROMIS-D, and SHAI questionnaires, assessing 'alexithymia', 'anxiety', 'depression', and 'health anxiety', were indeed most relevant for measuring their respective construct. Separate analyses were run for each measure. A Bayesian hierarchical model was fitted with *construct* as a fixed effect and *subject* and *item* as random effects.

*Second*, we examined whether the items of the TAS-20 subscales were most relevant for measuring 'alexithymia', compared to 'anxiety', 'depression', and 'health anxiety'. Separate analyses were run for each subscale. A Bayesian hierarchical model was fitted with *measure* as a fixed effect and *subject* and *item* as random effect. Additionally, we investigated whether the items from the TAS-20 subscales, assessing 'difficulty identifying feelings', 'difficulty describing feelings', and 'externally-oriented thinking', were most relevant for measuring the intended key features of alexithymia, i.e., 'difficulty identifying feelings', 'difficulty describing feelings', 'externally-oriented thinking', and 'limited imaginal capacity'.

*Finally*, a separate Bayesian hierarchical model was fitted for each single item of the TAS-20. The models included *construct* as a fixed effect and *subject* as a random effect. For all models described above, significance was evaluated at the 5% significance level (two-sided). Estimated mu parameters ($\hat{\mu}$) and their associated 95% credibility intervals (CI) are reported.

# RESULTS

## Participants

Data from 81 participants (63 females) were collected. After application of the manipulation checks (see section Detection of careless responding), data of 12 participants was removed from further analyses. More specifically, six participants failed to respond correctly to at least one of the IMC items, five participants provided unreliable data (i.e., at least one item was scored as −10 or +10 for all constructs), and one participant indicated that he/she was not attentive at all while completing the questionnaire. The final sample contained 69 participants (mean age of 21.07 years, $SD = 1.44$; 12 males). Most participants reported their ethnicity as Caucasian ($n = 61$). The large majority of participants (86%) reported to be mentally and physically healthy, 9% reported to be mentally troubled, 1% reported to be physically troubled, and 4% reported to be mentally and physically troubled. For the PROMIS, T-scores were 54.49 ($SD = 4.95$; range = 35.60–57.00) for physical function, 54.22 ($SD = 8.29$; range = 40.30–77.90) for anxiety, 50.36 ($SD = 8.26$; range = 41.00–79.40) for depression, 52.89 ($SD = 9.10$; range = 33.70–75.80) for fatigue, 48.22 ($SD = 7.92$; range = 32.00–68.80) for sleep

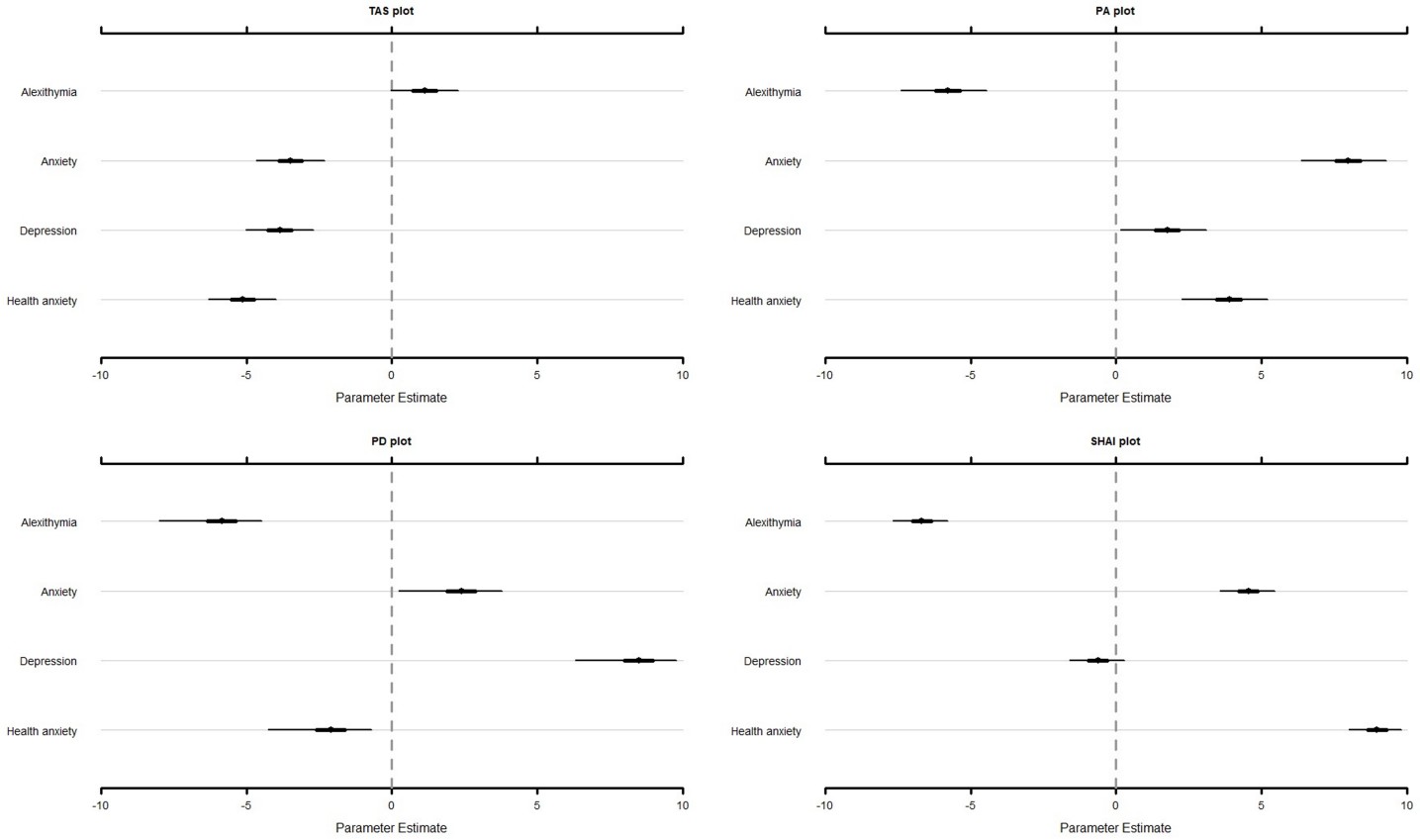

**Figure 1 Estimates and associated 95% credibility intervals of the relevance score for the TAS, PA, PD, and SHAI for alexithymia, anxiety, depression, and health anxiety.** TAS, Toronto Alexithymia Scale–20; PA, PROMIS® Item Bank v1.0-Emotional Distress-Anxiety–Short Form 4a; PD, PROMIS® Item Bank v1.0–Emotional Distress-Depression–Short Form 4a; SHAI, Short Health Anxiety Inventory.

disturbance, 53.75 ($SD$ = 7.41; range = 31.80–64.20) for ability to participate in social roles and activities, 46.62 ($SD$ = 7.47; range = 41.60–75.60) for pain interference. A mean score of 1.65 ($SD$ = 1.92; range = 0–8) was observed for pain intensity.

## Content validity of TAS-20 questionnaire and questionnaires of related constructs

### TAS-20 questionnaire

The items of the TAS-20 questionnaire scored significantly higher on 'alexithymia' ($\hat{\mu}$ = 1.12, 95% CI [−0.05 to 2.28]), compared to 'anxiety' ($\hat{\mu}$ =−3.48, 95% CI [−4.65 to −2.31]; Δ = 4.60, 95% CI [4.16–5.05]), 'depression' ($\hat{\mu}$ = −3.85, 95% CI [−5.03 to −2.68]; Δ = 4.98, 95% CI [4.53–5.42]), and 'health anxiety' ($\hat{\mu}$ = −5.13, 95% CI [−6.31 to −3.97]; Δ = 6.25, 95% CI [5.80–6.69]). It should be noted though that the score for 'alexithymia' was not significantly different from zero. Findings are displayed in Fig. 1.

### PROMIS-A questionnaire

The items of the PROMIS-A questionnaire scored significantly higher on 'anxiety' ($\hat{\mu}$ = 7.93, 95% CI [6.37–9.28]), compared to 'alexithymia' ($\hat{\mu}$ = −5.80, 95% CI [−7.39 to

−4.45]; Δ = 13.74, 95% CI [12.85–14.62]), 'depression' ($\hat{\mu}$ = 1.73, 95% CI [0.14–3.09]; Δ = 6.20, 95% CI [5.32–7.10]), and 'health anxiety' ($\hat{\mu}$ = 3.86, 95% CI [2.26–5.21]; Δ = 4.08, 95% CI [3.19–4.97]; Fig. 1).

### PROMIS-D questionnaire
Similar results were found for the PROMIS-D, showing that the items of the PROMIS-D scored significantly higher on 'depression' ($\hat{\mu}$ = 8.40, 95% CI [6.31–9.77]), compared to 'alexithymia' ($\hat{\mu}$ = −5.90, 95% CI [−8.00 to −4.46]; Δ = 14.30, 95% CI [13.45–15.16]), 'anxiety' ($\hat{\mu}$ = 2.33, 95% CI [0.24–3.78]; Δ = 6.07, 95% CI [5.20–6.93]), and 'health anxiety' ($\hat{\mu}$ = −2.15, 95% CI [−4.25 to −0.70]; Δ = 10.55, 95% CI [9.70–11.41]; Fig. 1).

### SHAI questionnaire
The items of the SHAI questionnaire scored significantly higher on 'health anxiety' ($\hat{\mu}$ = 8.95, 95% CI [7.98–9.81]), compared to 'alexithymia' ($\hat{\mu}$ = −6.69, 95% CI [−7.66 to −5.78]; Δ = 15.64, 95% CI [14.84–16.42]), 'anxiety' ($\hat{\mu}$ = 4.54, 95% CI [3.56–5.46]; Δ = 4.41, 95% CI [3.62–5.21]), and 'depression' ($\hat{\mu}$ = −0.63, 95% CI [−1.61 to 0.29]; Δ = 9.58, 95% CI [8.78–10.37]; Fig. 1).

## Content validity of the TAS-20 subscales
### Difficulty identifying feelings subscale
Analyses indicated that the items of the 'difficulty identifying feelings' subscale scored significantly higher on 'alexithymia' ($\hat{\mu}$ = 3.58, 95% CI [2.10–5.12]), compared to 'anxiety' ($\hat{\mu}$ = −1.56, 95% CI [−3.05 to −0.02]; Δ = 5.14, 95% CI [4.31–5.97]), 'depression' ($\hat{\mu}$ = −2.84, 95% CI [−4.32 to −1.30]; Δ = 6.42, 95% CI [5.60–7.25]), and 'health anxiety' ($\hat{\mu}$ = −2.16, 95% CI [−3.65 to −0.62]; Δ = 5.75, 95% CI [4.92–6.57]; Fig. 2A).

Furthermore, the items of the 'difficulty identifying feelings' subscale scored highest on 'difficulty identifying feelings' ($\hat{\mu}$ = 5.62, 95% CI [3.06–7.75]). Yet, compared to 'difficulty describing feelings', the difference was not significant ($\hat{\mu}$ = 2.70, 95% CI [0.15–4.81]; Δ = 2.92, 95% CI [2.18–3.65]). Furthermore, items of the 'difficulty identifying feelings' subscale scored significantly higher compared to 'externally-oriented thinking' ($\hat{\mu}$ = −4.53, 95% CI [−7.06 to −2.42]; Δ = 10.15, 95% CI [9.42 to 10.88]), and 'limited imaginal capacity' ($\hat{\mu}$ = −3.59, 95% CI [−6.15 to −1.48]; Δ = 9.21, 95% CI [8.48 to 9.94]; Fig. 2B).

### Difficulty describing feelings subscale
Analyses showed that the items of the 'difficulty describing feelings' subscale scored significantly higher on 'alexithymia' ($\hat{\mu}$ = 4.42, 95% CI [2.42–6.27]), compared to 'anxiety' ($\hat{\mu}$ = −4.08, 95% CI [−6.07 to −2.23]; Δ = 8.49, 95% CI [7.66–9.33]), 'depression' ($\hat{\mu}$ = −3.57, 95% CI [−5.56 to −1.72]; Δ = 7.99, 95% CI [7.15 to 8.83]), and 'health anxiety' ($\hat{\mu}$ = −6.84, 95% CI [−8.83 to −4.97]; Δ = 11.25, 95% CI [10.41–12.09]; Fig. 2A).

Furthermore, the items of the 'difficulty describing feelings' subscale scored significantly higher on 'difficulty describing feelings' ($\hat{\mu}$ = 6.72, 95% CI [4.56–8.65]), compared to 'difficulty identifying feelings' ($\hat{\mu}$ = 1.97, 95% CI [−0.19 to 3.91]; Δ = 4.74, 95% CI [3.82–5.67]), 'externally-oriented thinking' ($\hat{\mu}$ = −5.06, 95% CI [−7.20 to −3.14]; Δ = 11.78,

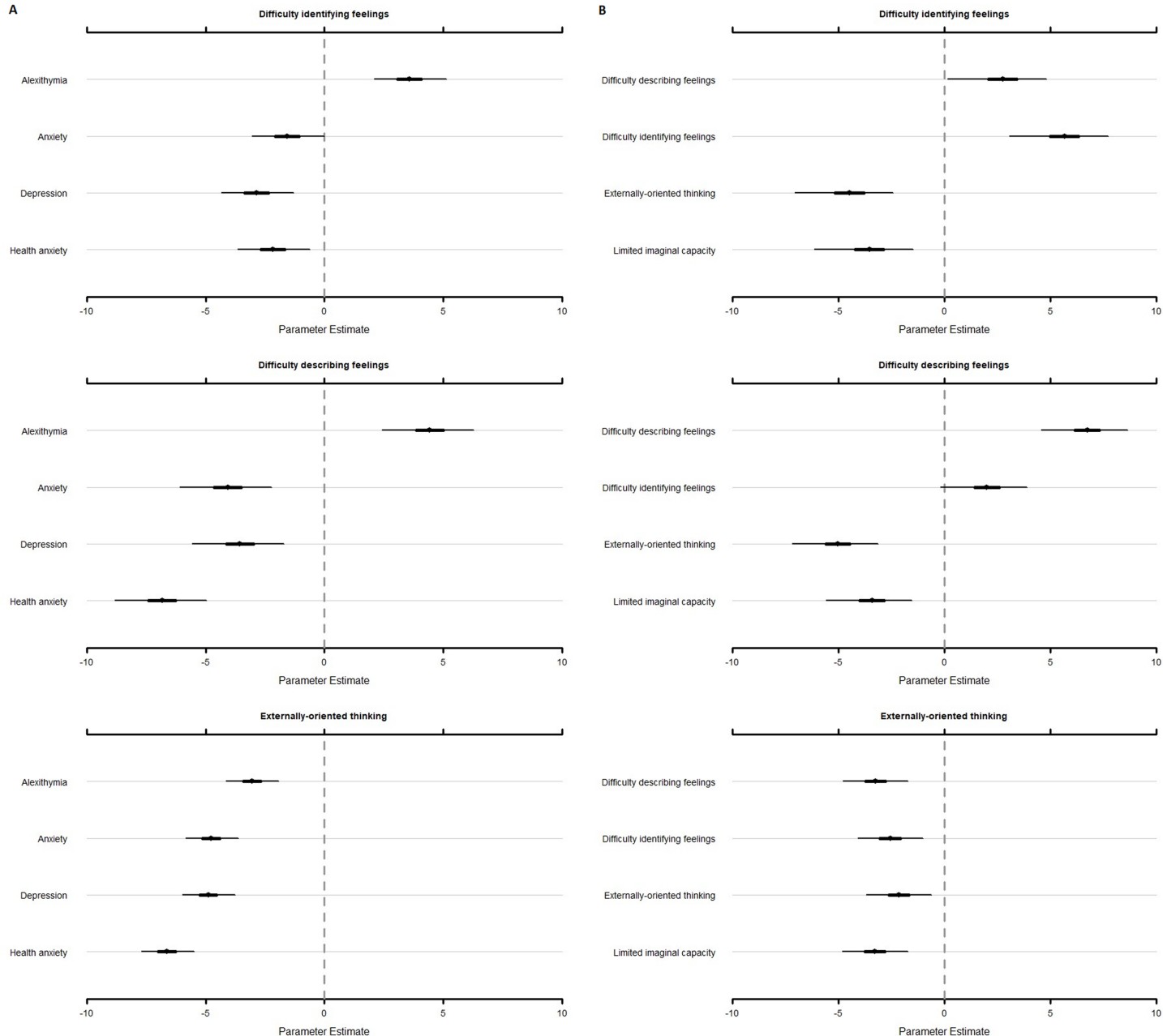

**Figure 2 Estimates and associated 95% credibility intervals of the relevance score for each TAS subscale on alexithymia, anxiety, depression, and health anxiety (A), and the alexithymia key features (B).** TAS, Toronto Alexithymia Scale–20; PA, PROMIS® Item Bank v1.0-Emotional Distress-Anxiety–Short Form 4a; PD, PROMIS® Item Bank v1.0–Emotional Distress-Depression–Short Form 4a; SHAI, Short Health Anxiety Inventory.

95% CI [10.86–12.70]), and 'limited imaginal capacity' ($\hat{\mu} = -3.45$, 95% CI [−5.59 to −1.53]; $\Delta = 10.16$, 95% CI [9.24 to 11.08]) (see Fig. 2B).

### Externally-oriented thinking subscale

The items of the 'externally-oriented thinking' subscale scored highest on 'alexithymia' ($\hat{\mu} = -3.05$, 95% CI [−4.13 to −1.93]). Yet, there is no significant difference compared to

'anxiety' ($\hat{\mu} = -4.76$, 95% CI [$-5.84$ to $-3.63$]) ($\Delta = 1.71$, 95% CI [$1.11$–$2.30$]), and 'depression' ($\hat{\mu} = -4.89$, 95% CI [$-5.97$ to $-3.76$]) ($\Delta = 1.83$, 95% CI [$1.24$–$2.43$]). 'Alexithymia' scored significantly higher compared to 'health anxiety' ($\hat{\mu} = -6.62$, 95% CI [$-7.70$ to $-5.49$]) ($\Delta = 3.57$, 95% CI [$2.97$–$4.16$]). However, note that $\hat{\mu}$ was negative for all constructs, indicating that the items of the 'externally-oriented thinking' subscale were not endorsed to measure 'alexithymia', nor 'anxiety', 'depression', or 'health anxiety' (Fig. 2A).

Furthermore, the items of the 'externally-oriented thinking' subscale scored highest on 'external oriented thinking' ($\hat{\mu} = -2.16$, 95% CI [$-3.69$ to $-0.62$]). Yet, no significant difference was found compared to 'difficulty describing feelings' ($\hat{\mu} = -3.26$, 95% CI [$-4.79$ to $-1.73$]; $\Delta = 1.10$, 95% CI [$0.37$–$1.82$]), 'difficulty identifying feelings' ($\hat{\mu} = -2.57$, 95% CI [$-4.09$ to $-1.02$]; $\Delta = 0.41$, 95% CI [$-0.31$ to $1.14$]), and 'limited imaginal capacity' ($\hat{\mu} = -3.28$, 95% CI [$-4.82$ to $-1.73$]; $\Delta = 1.13$, 95% CI [$0.41$–$1.85$]). Also here, note that $\hat{\mu}$ was negative for all constructs, indicating that the items of the 'externally-oriented thinking' subscale were not endorsed to measure 'externally-oriented thinking', nor 'difficulty describing feelings', 'difficulty identifying feelings', or 'limited imaginal capacity' (Fig. 2B).

## Content validity of TAS-20 items

### Difficulty identifying feelings items

The 'difficulty identifying feelings' subscale of the TAS-20 contains 7 items (items 1, 3, 6, 7, 9, 13, and 14). Results indicated that for all items, except item 3, $\hat{\mu}$ was positive and the confidence interval did not include 0, indicating that these items were endorsed to measure 'difficulty identifying feelings' (Fig. S1). Furthermore, item 6 ($\hat{\mu} = 8.36$, 95% CI [$6.99$–$9.66$]) and item 9 ($\hat{\mu} = 8.17$, 95% CI [$6.73$–$9.57$]) scored significantly higher on 'difficulty identifying feelings' than on all other constructs. For item 1, 13 and 14, the score on 'difficulty identifying feelings' (item 1: $\hat{\mu} = 7.08$, 95% CI [$5.68$–$8.49$]; item 13: $\hat{\mu} = 7.60$, 95% CI [$6.04$–$9.13$]; item 14: $\hat{\mu} = 5.66$, 95% CI [$4.16$–$7.16$]) was significantly higher for all constructs, except for 'difficulty describing feelings' (item 1: $\hat{\mu} = 4.41$, 95% CI [$3.01$–$5.82$]; item 13: $\hat{\mu} = 5.90$, 95% CI [$4.33$–$7.46$]; item 14: $\hat{\mu} = 3.32$, 95% CI [$1.83$–$4.81$]). For item 3, results indicated a significantly higher score on 'health anxiety' ($\hat{\mu} = 6.54$, 95% CI [$5.02$–$8.04$]) compared to all other constructs, including 'difficulty identifying feelings' ($\hat{\mu} = -1.74$, 95% CI [$-3.24$ to $-0.25$]). A significant positive score was also found for 'anxiety' ($\hat{\mu} = 1.88$ 95% CI [$0.39$–$3.38$]). Finally, item 7 scored significantly higher on 'difficulty identifying feelings' ($\hat{\mu} = 4.45$, 95% CI [$2.88$–$6.01$]), than on all other constructs, except for 'health anxiety' ($\hat{\mu} = 2.59$, 95% CI [$1.03$–$4.15$]; $\Delta = 1.85$, 95% CI [$-0.20$ to $3.92$]). A more detailed description and tabulation of the results is also provided in (Table S5).

### Difficulty describing feelings items

The 'difficulty describing feelings' subscale of the TAS-20 contains 5 items (items 2, 4, 11, 12, and 17). Results indicated that for all items, $\hat{\mu}$ was positive and the confidence interval did not include 0, indicating that these items were endorsed to measure 'difficulty describing feelings' (Fig. S2 and Table S5). In addition, item 2 ($\hat{\mu} = 8.42$, 95% CI

[7.04–9.70]), item 11 ($\hat{\mu}$ = 8.04, 95% CI [6.68–9.37]), item 12 ($\hat{\mu}$ = 6.49, 95% CI [4.99–7.99]) and item 17 ($\hat{\mu}$ = 6.72, 95% CI [5.22–8.24]) scored significantly higher on 'difficulty describing feelings' than on all other constructs. For item 4, the score on 'difficulty describing feelings' ($\hat{\mu}$ = 4.29, 95% CI [2.89–5.69]) was significantly higher than for all other constructs, except for 'difficulty identifying feelings' ($\hat{\mu}$ = 2.32, 95% CI [0.92–3.75]; $\Delta$ = 1.97, 95% CI [0.14–3.80]).

### Externally-oriented thinking items

The 'externally-oriented thinking' subscale of the TAS-20 contains 8 items (items 5, 8, 10, 15, 16, 18, 19, and 20). Results indicated that for item 8 and item 20, $\hat{\mu}$ was positive and the confidence interval did not include zero, indicating that these items were endorsed to measure 'externally-oriented thinking' (Fig. S3 and Table S5). For both items, the score on 'externally-oriented thinking' (item 8: $\hat{\mu}$ = 1.48, 95% CI [0.06–2.91]; item 20: $\hat{\mu}$ = 1.45, 95% CI [0.09–2.79]) was significantly higher than on all other constructs. For item 15, results indicated that $\hat{\mu}$ was positive but the confidence interval did include zero, indicating that this item was not endorsed to measure 'externally-oriented thinking'. Item 15 was however endorsed to measure 'difficulty describing feelings' ($\hat{\mu}$ = 2.38, 95% CI [0.88–3.86]). For all other items (item 5: $\hat{\mu}$ = −5.01, 95% CI [−6.22 to −3.79]; item10: $\hat{\mu}$ = −3.86, 95% CI [−5.28 to −2.44]; item 16: $\hat{\mu}$ = −1.15, 95% CI [−2.66 to 0.36]; item 18: $\hat{\mu}$ = −5.41, 95% CI [−6.75 to −4.08]; item 19: $\hat{\mu}$ = −4.92, 95% CI [−6.19 to −3.64]), $\hat{\mu}$ was negative, indicating that these items were not endorsed to measure 'externally-oriented thinking'. For these items, also all other constructs were non-significant or significantly negative, showing that the items were not endorsed to measure any of these constructs either.

## DISCUSSION

The present study investigated the content and discriminant content validity of the TAS-20, a widely used self-report measure of alexithymia (*Bagby, Parker & Taylor, 2020*). Using the Discriminant Content Validity method (DCV; *Johnston et al., 2014*), participants rated the extent to which each TAS-20 item was relevant for measuring 'alexithymia' and its key features (content validity), or related constructs, i.e., 'anxiety', 'depression', and 'health anxiety' (discriminant content validity). The results can be readily summarized. First, results showed that participants did not endorse the TAS-20 as measuring 'alexithymia', whereas the PROMIS-A, PROMIS-D, and SHAI did distinctively measure their intended construct. Second, the subscales 'difficulty identifying feelings' and 'difficulty describing feelings' measure 'alexithymia'. This was not the case for the 'externally-oriented thinking' subscale. Additionally, results indicated that the 'difficulty describing feelings' subscale distinctively assessed its intended construct. This was not the case for the 'difficulty identifying feelings' subscale, which measures both 'difficulty identifying feelings' and 'difficulty describing feelings'. Furthermore, the 'externally-oriented thinking' subscale assessed none of the included constructs. Finally, results showed that eight items distinctively measured their intended construct, and four items measured both the 'difficulty identifying feelings' and the 'difficulty describing feelings'

constructs. Two items assessed (health) anxiety equally well or even better, and none of the items (except item 3) showed content overlap with 'anxiety' or 'depression'.

To our knowledge, this study is the first to empirically investigate the content of the TAS-20. Until now content validity has been largely overlooked at the expense of other forms of validity, such as construct (i.e., convergent and discriminant validity) and criterion validity (i.e., predictive, concurrent, and retrospective validity) (e.g., *Lumley, Neely & Burger, 2007*; *Parker, Taylor & Bagby, 2003*; *Bagby, Parker & Taylor, 2020*). This is surprising as content validity is a fundamental property of any measure of any theoretical construct (*Haynes et al., 1995*) and key in theory testing, intervention design, and practical applications (*Dixon & Johnston, 2019*; *Van Ryckeghem, 2021*).

The results of the current study call for reflection. First, overall, the TAS-20 was not considered relevant for measuring 'alexithymia'. Furthermore, the 'difficulty identifying feelings' subscale assessed both the 'difficulty identifying feelings' and the 'difficulty describing feelings' constructs, and the 'externally-oriented thinking' scale was not identified as measuring 'externally-oriented thinking', nor 'alexithymia'. These findings put a threat on the interpretation of earlier and future studies using the TAS-20 as findings are potentially flawed due to a lack of content validity. Nonetheless, the finding that only the 'difficulty identifying feelings' subscale and the 'difficulty describing feelings' subscale were content valid for 'alexithymia' is not surprising. *Sifneos (1973)* (p. 256) stated that "for lack of a better term", he proposed the term "alexithymic" (from Greek stems a = lack, lexis = word, and thymos = mood or emotion) to denote "the most striking characteristic", namely the inability of these patients to find appropriate words to describe their feelings (*Nemiah & Sifneos, 1970*). Although the literal meaning of the term alexithymia-'without words for feelings'-refers to this particular characteristic (*Apfel & Sifneos, 1979*), Sifneos made repeatedly clear that the term 'alexithymia' is the name of a construct that encompasses multiple characteristics (e.g., *Nemiah, Freyberger & Sifneos, 1976*; *Sifneos, 1994*, *1996*). Therefore, to define the alexithymia construct in our study, we chose not to use the literal meaning but instead turn to the definition that is used in scientific literature (*Taylor, Bagby & Parker, 2016*) and understood in lay terms (online Oxford Living Dictionaries for English)—'The inability to recognize one's own emotions and to express them, especially in words.' Although this definition has a broad scope, its focus is on only two out of the four key features, namely the 'difficulty identifying feelings' feature referring to the inability to recognize and the 'difficulty describing feelings' feature referring to the inability to express. In line with this reasoning, we see a plausible explanation for the finding that the 'externally-oriented thinking' subscale was not identified to measure the alexithymia construct. However, by including each of the definitions of the key features, we expected that the 'externally-oriented thinking' subscale would be identified as a measure of the externally-oriented thinking construct, and potentially as an indirect measure of the limited imaginal capacity construct. This was not the case. Analyses of the individual externally-oriented thinking items corroborated this finding. Only two items of this scale were perceived as measuring externally-oriented

thinking and none as measuring limited imaginal capacity. Closer inspection of these results showed that four of the items that did not perform well, were reverse-scored items. However, this is not a full explanation because the two other items were regularly keyed. This observation is in line with the findings of Preece et al. (2018a), who performed confirmatory factor analysis and showed the presence of a reverse-scored item method factor, but also poor factor loadings for the regular-keyed items. Our findings are important as they signal the need of revising the items designed to measure externally-oriented thinking so that they represent their intended construct more accurately. Furthermore, our findings contradict Bagby, Parker & Taylor (1994a, 1994b) assumption on the representation of the limited imaginal capacity feature in externally-oriented thinking items. The present results indicate that caution is warranted in using the TAS-20 in its entirety: only two out of the four key features of the alexithymia construct are represented in the item pool.

We also observed that multiple difficulty identifying feelings items had higher scores on the 'difficulty identifying feelings' construct than on the 'difficulty describing feelings' construct. Furthermore, one difficulty describing feelings item had higher scores on the 'difficulty describing feelings' construct than on the 'difficulty identifying feelings' construct. These findings are in line with the results of various studies, revealing that difficulty identifying feelings items and difficulty describing feelings items are closely related, and subscale scores often correlate highly (e.g., $r = 0.43–0.80$; Kooiman, Spinhoven & Trijsburg, 2002). Current findings suggest that content overlap between both subscales may (at least partly) be the basis of observed correlations. Together with the fact that some studies showed that the items of these subscales merge into one single factor (e.g., Erni, Lötscher & Modestin, 1997; Loas et al., 1996), current findings provide support for the idea that part of the items on these scales probably represent the same aspect of alexithymia (Kooiman, Spinhoven & Trijsburg, 2002, but see Gignac, Palmer & Stough, 2007). Future research is needed to examine why the wording and phrasing of some of these items is perceived as measuring both constructs.

Finally, two TAS-20 items of the 'difficulty identifying feelings' scale that are developed to measure the difficulty in differentiating between bodily feelings and emotions showed to measure other constructs. One TAS-20 item was identified to measure 'anxiety' and 'health anxiety' (i.e., "I have physical sensations that even doctors don't understand."), the other item was identified to measure both 'health anxiety' and 'difficulty identifying feelings' (i.e., "I am often puzzled by sensations in my body"). Due to this content overlap, these TAS-20 items may result in a misleading evaluation of patients suffering from medically unexplained symptoms. A total score on the TAS-20 may overestimate the prevalence or severity of alexithymia in patients with medical conditions. To avoid unduly psychologization, caution is warranted in interpreting the TAS-20 overall score as a straightforward measure of alexithymia in these populations. One possibility would be to make a separate scale of these items allowing to check/control for their contribution in the TAS-20 total score. A recent study of Fournier et al. (2019) provides a potential starting

point. In line, with current findings they found that both item 3 and item 7 form a new latent factor *difficulty in interoceptive abilities* that is specifically related to health and personality trait outcomes. Alternatively, current findings may indicate that a standalone use of this self-report questionnaire may be insufficient to assess alexithymia. There is a need to take into account the context. Indeed, the use of a multimethod approach including the TAS-20, accompanied by a clinical interview and expert judgement may be required to assess whether elevation of self-reported alexithymia levels can be attributed to different aspects of the presentation of alexithymia, the context of the person, or comorbid medical conditions (see also *Bagby et al., 2006*; *Taylor, Bagby & Parker, 2016*). Future research is warranted to further scrutinize this topic.

Finally, the TAS-20 items did show discriminant content validity with 'anxiety' (except item 3) and 'depression'. This supports the idea that the TAS-20 is not merely a measure of negative affect (*Lumley, 2000*; *Bagby, Parker & Taylor, 2020*), indicating that high correlations between alexithymia and anxiety/depression are not due to content overlap between scales of both constructs. Yet, it remains possible that the high correlations between alexithymia and negative affect are due to the particular formulation of a substantial part of the TAS-20 items. Indeed, a substantial number of items of the 'difficulty identifying feelings' subscale and the 'difficulty describing feelings' subscale are negatively phrased (e.g., "I find it hard to describe how I feel about people"). It is known that people high in negative affectivity tend to manifest a general tendency towards a self-effacing response style or self-criticism, thus, tend to report negative things about themselves on self-report questionnaires generally (*Lumley, 2000*).

This study has some limitations. First, healthy lay people, and no experts or patients were involved. The nature of the discriminant validity method is designed to allow lay people without scientific background (and thus knowledge biases) to judge whether items assess a certain construct (see also *Crombez et al., 2020*). However, no agreement exists whether experts should be used who are familiar with the theoretical constructs, or whether non-biased lay people should be used who are the putative respondents of the measure (*Dixon & Johnston, 2019*). Second, judges were mainly female, which precluded the examination of gender effects. Third, the DCV method provides a quantitative analysis of content validity. Other methods are possible, and may provide insight in how participants mentally process and respond to items. One promising procedure to provide a qualitative analysis of content validity is cognitive interviewing (*Willis, 2015*). Fourth, the statistical analyses differed from the analyses performed in typical research with the DCV method (*Johnston et al., 2014*, but see *Crombez et al., 2020*). These studies generally involve a low number of participants (at least 15 participants are recommended) and use primarily one-sample or paired t-tests. Hence, results then strongly depend on sample size and statistical power. The analyses in the present research were performed using Bayesian hierarchical models, which has several advantages (see earlier), but also requires a larger number of participants. Therefore, the sample size of current study largely exceeds previous studies resulting in narrow confidence intervals. Fifth, we have only included the TAS-20. Other measures exist such as the Bermond Vorst Alexithymia Questionnaire (BVAQ; *Vorst & Bermond, 2001*), the Psychological Treatment

Inventory-Alexithymia Scale (PTI-AS; *Gori et al., 2012*), and the Perth Alexithymia Questionnaire (PAQ; *Preece et al., 2018b*). Although promising results on the psychometric properties of the latter measures have been reported (*de Vroege et al., 2018*; *Bagby et al., 2009*; *Vorst & Bermond, 2001*), future research is needed to corroborate the (discriminant) content validity of these questionnaires as well.

## CONCLUSIONS

The TAS-20, currently the most utilized instrument to assess alexithymia and its key features is found to be only partially content valid. Particularly, current findings indicate problems with the content validity of the TAS-20 questionnaire, and some of its subscales do not measure the intended key features. Indeed, only the subscales 'difficulty identifying feelings' and 'difficulty describing feelings' represented 'alexithymia' and their intended construct. This was not the case for the 'externally-oriented thinking' subscale, which assessed none of the alexithymia key features or related constructs. Finally, some items of the TAS-20 are contaminated with content measuring (health) anxiety. Due to described problems with (discriminant) content validity, revision of the TAS-20 is recommended to adequately assess (all key features of) alexithymia. Furthermore, caution is warranted when assessing the TAS-20 in people suffering from medical conditions.

## ACKNOWLEDGEMENTS

We thank Clara Coen, Elena Dhondt, Thea Herfs, Arawa Kolossa, Lynn Pasch, Glenn Proctor, Marie Santillo, and Dyonne Vrouenraets for their help in the data collection.

### Funding

The authors received no funding for this work.

### Competing Interests

The authors declare that they have no competing interests.

### Author Contributions

- Elke Veirman conceived and designed the experiments, authored or reviewed drafts of the paper, and approved the final draft.
- Dimitri M.L. Van Ryckeghem conceived and designed the experiments, performed the experiments, authored or reviewed drafts of the paper, and approved the final draft.
- Gregory Verleysen analyzed the data, prepared figures and/or tables, and approved the final draft.
- Annick L. De Paepe analyzed the data, authored or reviewed drafts of the paper, and approved the final draft.
- Geert Crombez conceived and designed the experiments, authored or reviewed drafts of the paper, and approved the final draft.

## Human Ethics

The following information was supplied relating to ethical approvals (i.e., approving body and any reference numbers):

The Ethics Review Committee Psychology and Neuroscience (ERCPN) of Maastricht University granted ethical approval to carry out the study within its facilities (Ethical Application Ref: RP2027_2019_16).

## Data Availability

The data and R script are available available at OSF: Veirman, Elke, Van Ryckeghem Dimitri, Gregory Verleysen, Annick De Paepe, and Geert Crombez. 2021. "A Discriminant Content Validity Study of the Toronto Alexithymia Scale-20." OSF. January 3. DOI 10.17605/OSF.IO/4F5SE.

## Supplemental Information

Supplemental information for this article can be found online at http://dx.doi.org/10.7717/peerj.11639#supplemental-information.

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
