# Peer review of "What do alexithymia items measure? A discriminant content validity study of the Toronto-alexithymia-scale–20"

_PeerJ, doi:10.7717/peerj.11639_

## Round 0.1 · original submission · Major Revisions

· Academic Editor

Major Revisions

Thank you for submitting your manuscript to PeerJ. I have now received two reviews, and I would like to thank both reviewers for their thoughtful feedback on the paper. Both reviewers see merit in the manuscript; however, both also have a number of comments that require addressing. The reviews are appended below, so I will not reiterate all the comments here; however, please note that Reviewer 2 was sent Tables 1 and 2 (and has no additional comments).

I do think the comments regarding the analyses (Reviewer 1) and sampling (Reviewer 1 and Reviewer 2) warrant particular attention. I also think Reviewer 2’s suggestion to engage more with alternative assessments of alexithymia (both self-report measures as well as observer ratings) is a good one, and that discussing this would strengthen the manuscript. That said, please address all the reviewer comments in your response.

Thank you for again submitting your manuscript to PeerJ and I hope you find these comments, along with the reviews, useful in revising your manuscript.

·

Basic reporting

- Good theoretical justification of the study.
- The issue of discriminant and content validity of the TAS-20 is clearly explained
- Overall, very good justification of the study.

Experimental design

- In the "Identification of construct" section, the use of the "Other" category was a little confusing. The section appears to mix design and results (including it in the classification but then excluding it). Please make this more clear.
- Description of the procedure was clear and followed a logical pattern.
- In terms of analyses, I have no expertise in Bayesian hierarchical models, so I don't feel comfortable commenting on the analysis and specifics of the results.
-Another way of analysing the data to test discriminant validity would be to conduct an EFA where all items are entered and the expectation would be that they load on the intended factors (n this case, the respective the scales). Although I'm not in a position to say that the Bayesian option was not appropriate, I wonder if the authors could comment more (apart from the single line provided "In line with previous research ") on the choice of analytical strategy.

Validity of the findings

- I agree with the authors on the idea that examination of the content validity of the TAS-20 has been largely overlooked
- Nothing was done with the observation of reverse scored items, which has been found to be problematic in terms of the scale's validity.
- Nice to see corroborated, from this perspective, the well documented problems with the EOT subscale.
- The conclusion "The present results suggest caution in using the TAS-20 in its entirety as mainly two out of the four key features of the alexithymia construct are represented in the items" is measured and balanced.
- Many of the claims in this paper are well supported with relevant literature.
- That some items overlap with ‘health anxiety’ is a shrewd observation. I'd be careful though in overinterpreting this in that many psychological scales (e.g., those assessing depression) would overlap with medical conditions (e.g., low sugar in diabetes). Clinical judgement, considering context, allows attributing findings to different aspects of the presentation.
- This comment in the conclusions "The TAS-20....is not endorsed to measure alexithymia..." is too strong. No single study can claim this. I partly agree with the findings, but the comment is an overstatement.
- Although the authors acknowledge this point in the limitations section of the submission, using lay people to be the judge of certain psychological (or any for that matter) disorder/presentation is a bit of a problematic assumption. A crude extension would be a medical condition and the opinion of the person with the ailment. Or asking people with panic disorder if their so called intuitions" are a true reflection of a coming physical episode. Anyway, more on this should have been presented in the intro as a way of justifying this methodology.

·

Basic reporting

no comment, well described, the use of English is professionaly sound.

Experimental design

no comment, solid design. Only limitation in my point of view is the limited number of participants. Please adress this is in the discussion.

Validity of the findings

The authors adress the presence of other measures such as the BVAQ and the TI-AS in one sentence in the discussion. I would like to advise the authors to elaborate on this. The authors correctly state that the TAS-20 has its flaws considering content validity. On the contrary, the BVAQ (with another conceptualisation of alexithymia) looks promoising (see de Vroege et al. 2018; Bagby et al., 2009; Vorst & Bermond, 2001). What is the authors' view on these implications for the construct of alexithymia?

Furthermore, the use of self-report questionnaires like the TAS-20 and BVAQ may be problematic in assessing alexithymia; see Bagby et al., 2006. Please include this in your section about future studies.

Suggested references

Bagby RM, Taylor GJ, Parker JD, Dickens SE. The development of the Toronto
Structured Interview for Alexithymia: item selection, factor structure, reliability and concurrent validity. Psychother Psychosom (2006) 75(1):25–39. doi:10.1159/
000089224

Bagby RM, Quilty LC, Taylor GJ, Grabe HJ, Luminet O, Verissimo R, et al.
Are there subtypes of alexithymia? Pers Individ Dif (2009) 47(5):413–8.
doi:10.1016/j.paid.2009.04.012

Vorst HC, Bermond B. Validity and reliability of the Bermond-Vorst
Alexithymia Questionnaire. Pers Individ Dif (2001) 30:413–34. doi:10.1016/
S0191-8869(00)00033-7

de Vroege, L., Emons, W. H. M., Sijtsma, K., & van der Feltz-Cornelis, C. M. (2018). Psychometric properties of the Bermond-Vorst Alexithymia Questionnaire (BVAQ) in the general population and a clinical population. Frontiers in Psychiatry, 9, [111]. https://doi.org/10.3389/fpsyt.2018.00111

Additional comments

Major point;
table 1 and table 2 are missing in the review file that I downloaded. I cannot advise acceptation of the manuscript before reviewing these two tables. Nevertheless, Vierman and colleagues adressed an interesting topic and explored the content validity of the TAS-20 in an elegant way.

---

## Round 0.2 · accepted · Accept

· Academic Editor

Accept

Thank you for addressing the comments provided the reviewers on the original manuscript. I am delighted to be able to accept your paper for publication in PeerJ.

·

Basic reporting

no other comments, the authors adressed my concerns.

Experimental design

no other comments, the authors adressed my concerns.

Validity of the findings

no other comments, the authors adressed my concerns

Additional comments

I would be happy to advise the editor to accept the manuscript in its current form for publication. I congratualte the authors with an interesting paper and study.

Best regards,
Lars de Vroege